# An Old Acquaintance: Could Adenoviruses Be Our Next Pandemic Threat?

**DOI:** 10.3390/v15020330

**Published:** 2023-01-24

**Authors:** Gustavo Saint-Pierre Contreras, Daniel Conei Valencia, Luis Lizama, Daniela Vargas Zuñiga, Luis Fidel Avendaño Carvajal, Sandra Ampuero Llanos

**Affiliations:** 1Programa de Virología, ICBM, Facultad de Medicina, Universidad de Chile, Independencia 1027, Santiago 8380453, Chile; 2Unidad Microbiología, Hospital Barros Luco Trudeau, Servicio de Salud Metropolitano Sur, Santiago 8900000, Chile; 3Departamento de Ciencias de la Salud, Universidad de Aysén, Coyhaique 5951537, Chile

**Keywords:** infections, human adenovirus, social determinants of health, adenovirus vaccine, genotype, molecular diagnostic testing

## Abstract

Human adenoviruses (HAdV) are one of the most important pathogens detected in acute respiratory diseases in pediatrics and immunocompromised patients. In 1953, Wallace Rowe described it for the first time in oropharyngeal lymphatic tissue. To date, more than 110 types of HAdV have been described, with different cellular tropisms. They can cause respiratory and gastrointestinal symptoms, even urinary tract inflammation, although most infections are asymptomatic. However, there is a population at risk that can develop serious and even lethal conditions. These viruses have a double-stranded DNA genome, 25–48 kbp, 90 nm in diameter, without a mantle, are stable in the environment, and resistant to fat-soluble detergents. Currently the diagnosis is made with lateral flow immunochromatography or molecular biology through a polymerase chain reaction. This review aimed to highlight the HAdV variability and the pandemic potential that a HAdV3 and 7 recombinant could have considering the aggressive outbreaks produced in health facilities. Herein, we described the characteristics of HAdV, from the infection to treatment, vaccine development, and the evaluation of the social determinants of health associated with HAdV, suggesting the necessary measures for future sanitary control to prevent disasters such as the SARS-CoV-2 pandemic, with an emphasis on the use of recombinant AdV vaccines to control other potential pandemics.

## 1. Introduction

Human adenovirus (HAdV) and respiratory syncytial virus (RSV) are recognized pathogens in pediatric and elderly patients, generating significant public health expenditures annually. In 1953, Wallace Rowe described it for the first time in oropharyngeal lymphatic tissue [1,2]. Subsequently, HAdV showed as a virus that produces malignant tumors in newborn rodents. Several studies on the pathogenesis and the replicative cycle of HAdV demonstrated the relationship between viral replication and oncogenesis in rodents [3,4,5,6]. HAdV was eventually shown to have no role in human cancer [7]. The results derived from these investigations have had a great impact to understand the expression of adenovirus genes in mammals and its use as a platform in the development of vaccines [1,8]. Unlike the respiratory syncytial virus and influenza viruses, which have a clear seasonal predominance, respiratory HAdV generates outbreaks continuously throughout the year, as the SARS-CoV-2 pandemic has behaved up to now [1,9,10].

HAdV is associated with a wide range of diseases ranging from mild respiratory conditions such as pharyngitis or conjunctivitis to more serious conditions such as gastroenteritis with dehydration, severe acute respiratory infections, hemorrhagic cystitis, and meningoencephalitis [11,12,13]. HAdV can cause severe and fatal disease in both immunocompetent and immunocompromised hosts, particularly in the pediatric population under 6 years of age. HAdV causes 5% to 10% of all febrile illnesses in infants and preschool-age patients. The infection is usually self-limiting, but persistent viral excretion has been described for 4 weeks, even up to 18 months in the preschool population and immunosuppressed patients [14,15]. In some studies, scientists have postulated that this mechanism would be the source of endemic circulation and sometimes epidemic outbreaks in certain closed populations [9,16]. A high prevalence of HAdV has been described in some populations, particularly type 5, one of the most studied in humans. Zhang et al. in 2013, detected a 73.1% seroprevalence in the adult population from various regions of China [17]. Other studies have described the same serological evidence in the pediatric population; for example, in the Chengxi district, children between 4 and 7 years old have a HAdV-5 seroprevalence of 73% [1,18,19,20,21]. In the US, in 1963, the adult female population showed a 71% prevalence for HAdV-4 in Washington D.C., and in other states, there was a range between 12% and 29% [22].

Transmission of HAdV in swimming pools by fecal–oral mechanisms has been documented in the literature, in waters contaminated with stools from patients carrying HAdV, mainly in non-chlorinated swimming pools [23]. It has been shown that HAdV can be transmitted between people with poor hand hygiene since this virus can replicate in epithelial cells of the gastrointestinal and respiratory tracts, ocular conjunctiva, and even in the bladder [9,24].

HAdVs are double-stranded DNA viruses, are highly stable in the environment, nonenveloped viruses, and resistant to fat-soluble detergents [9,24]. The spread of HAdV through local spread (outbreaks) in orphanages, daycare centers, schools, and even in summer camps or closed communities such as military camps and prisons is characteristic [9,16,25,26]. Among immunocompromised patients, infection is most frequently described in hematopoietic stem cell transplant recipients (HSCT) and solid organ transplant (SOT) recipients [27]. HAdV infection continues to be a prevalent disease worldwide; the admission of patients to the critical care (ICU) has been described, including death in specific populations and pediatric and immunocompromised patients. The main focus of this review was to recognize the role of adenoviruses as a potential producer of future pandemics during the 21st century, as well as their use as viral vector vaccines in the control of other pandemics.

## 2. History of HAdVs

In 1953, Wallace Rowe and collaborators observed that the fluid secreted by the adenoidal tissue (lymph nodes and palatine tonsils) in the upper airway, when inoculated into HeLa cell cultures, degenerated spontaneously. They observed clusters of rounded cells followed by cytoplasmic inclusions [28]. As a consequence, this group identified a new virus that was called Adenovirus, due to the adenoidal tissue where it was found [1,28]. They were unaware of the pathogenic role of HAdV, even postulating in their article: “Further research is underway to determine the agent’s relationship with adenoids and to study its possible role in human disease; particularly upper respiratory tract infections” [28]. In 1955, Berge et al. would be the first to describe HAdV as the cause of respiratory disease in humans, particularly in military personnel in the US [29].

In 1962, HAdVs were the first human viruses to be associated with the development of cancer in other species, but this has never been demonstrated in humans. These studies were originally carried out in newborn hamsters but were later replicated in other rodent models. Finally, they were also replicated in baboons [7,30]. Subsequently, human papillomaviruses (HPV), polyomaviruses, and adenoviruses were collectively classified as small-DNA tumor viruses, based on their small double-stranded DNA genome sizes and their ability to induce cancer in experimental systems or animal models and humans [7]. During the same year, 1962, at the VIII International Congress of Microbiology, the virology subcommittee, dependent on the international nomenclature committee, decided to classify Adenoviruses based on criteria previously established by Rowe and his collaborators in 1955. Among others, these include the resistance to ether, the behavior of HAdV in cell culture lines, and specific soluble antigens for adenovirus groups, generating six subgroups according to the affected species (human, simian, bovine, canine, murine, avian). These subgroups were later reclassified as species maintaining currently recognized alphabetical order [31]. To date, neither widely available vaccines nor approved antiadenoviral compounds are available to efficiently treat HAdV infections. Thus, there is a need to thoroughly understand HAdV-induced disease, and for the development and preclinical evaluation of HAdV therapeutics and/or vaccines, and consequently for suitable and standardizable in vitro systems and animal models such as hamsters, mice, and rabbits [32].

## 3. HAdV Classification (Types)

According to the latest update of the International Committee on Viral Taxonomy (ICTV) on the virus taxonomy profile of the year 2022, the demarcation of genus and species is based mainly on phylogenetic criteria, but also on the organization of the genome and the biological characteristics within the genera are described as follows: Genus Mastadenovirus: >500 types (members that infect mammals); Aviadenovirus: >14 types (infection in birds); Atadenovirus: >9 types (reptiles, birds, ruminants, and marsupials); Siadenovirus: >7 types (birds, frogs, and turtles); Ichtadenovirus: 1 type (white sturgeon); Testadenovirus: 1 type (red-eared slider turtle) [33]. HAdVs are members of the family *Adenoviridae* and Genus Mastadenovirus. There are currently more than 110 types of HAdV reported [24,33,34], but the pathogenicity of many of them is unknown [9,24]. HAdVs are classified into seven species (A–G), with multiple types in each of them [35].

## 4. Virological Characteristics of HAdV and Its Pathogenesis

HAdV, linear double-stranded DNA, 25–48 kbp in length, are very stable in the environment, do not have a lipid envelope, and are less vulnerable to lipid-soluble detergents [9,24]. HAdV can remain for weeks in the environment and when frozen at −20 °C can survive for years, which generates a comparative advantage over other viruses to remain stable on surfaces, which is also advantageous for researchers to carry out studies with samples frozen for years [9,36].

Other recognized morphological characteristics of the virion are being 90 nm in diameter, having an icosahedral-type capsid, with 240 capsomeres without a vertex (hexons) and 12 capsomeres with a vertex (pentons); the latter fibers stand out, which are homotrimers of protein IV, consisting of three structural domains: the tail, which is attached to the base of the penton, the axis of characteristic length, and the distal bulge [24,33] (Figure 1A,B).

The adenoviral genome (Figure 2) includes four early genes (E1–E4) and five late genes (L1–L5) that are transcribed before and after viral DNA replication, respectively [37,38]. Early genes encode proteins that activate transcription of other viral genes and mediate viral DNA replication, while late genes encode primarily viral structural proteins [38]. Immediate early gene (E1A) is expressed for the first time after HAdV infection and is the most important transcriptional activator for subsequent viral gene expression. The critical adenovirus factors involved in the helper effect are the products of the E1a, E1b, E4 (orf6), and E2a genes. Late genes encode structural/capsid viral proteins (penton, hexon, and fiber) and core viral protein (e.g., protein VII and protease). Finally, at both ends of the genome, there are inverted terminal repeats (ITR) of 145 bases in length, which flank two open reading frames (ORF), E1 and E4. The ITRs constitute the viral sequences required in cis for DNA replication and encapsidation [39].

Regarding the replicative cycle, HAdV causes a lytic infection in epithelial cells, being the entry gate cells of the upper respiratory epithelium, such as tonsils and lymphoid tissue; at the same time, a cycle of latent infection in lymphoid cells of the gastrointestinal tract has been recognized [9,40,41].

This virus is transmitted by air through droplets, but also by aerosols [42,43], transmission by sneezing or coughing being the most frequently recognized; however, transmission by the fecal–oral route and/or direct contact with secretions has also been demonstrated. Its spread is recognized in the literature through local outbreaks [44] or closed communities as previously mentioned [16,25,26]. It can also generate deadly cases in intra-hospital outbreaks, for which isolation in an individual room is required, with ventilation to the outside, use of contact precautions and additional precautions described by the World Health Organization (WHO) and the Centers for Disease Control and Prevention (CDC), to avoid contagion to other patients or healthcare workers [24,45,46,47,48]. In addition, it is recommended not to isolate in a cohort with other HAdV cases due to the possibility of cross-contagion with other genotypes, including genetic recombination between them, as occurred with HAdV-D53, a recombinant of other HAdVs previously known to cause viral keratoconjunctivitis [9,49].

## 5. Microbiological Diagnosis

The most frequent sample analyzed for the study of respiratory viruses is naso-oropharyngeal secretion obtained through swabbing or secretion aspiration. Before the (H1N1)pdm09 influenza virus pandemic (2009), the etiological diagnosis of respiratory viruses was made with desquamated cells from the upper airway by direct immunofluorescence (IF) [50]. This technique uses specific antibodies against viral antigens of the HAdV species expressed in epithelial cells and marked with fluorescent dyes whose positivity is analyzed through fluorescence microscopy [51,52,53]. Although this technique is still used in healthcare centers, many have begun to apply molecular biology techniques for its diagnosis, due to the low sensitivity of the direct IF test in the clinical setting 41.7% (95% specificity) [54].

On the other hand, for some authors, the enzyme immunoassay (EIA) is the method of choice for the analysis of soluble adenovirus antigens in feces and respiratory secretions. These assays may be directed against a common hexon antigen, being capable of detecting any known adenovirus serotype, or simply a serotype-specific antigen. However, genotypic variants could not be diagnosed or classified by this technique. Specificity and sensitivity are 90%–95% and 60%–85% respectively compared to the most complex and complete isolation techniques in cell culture [55,56]. A method that combines EIA with lateral flow immunochromatography has recently appeared and seems to be more sensitive and specific, with results in 10 min (72% sensitivity and 100% specificity when compared to cell culture). Despite the good results, an argument against it is its high costs compared with other techniques. Currently, in some countries it is more convenient to perform real time-PCR on Multiplex Platforms for the Identification of Respiratory pathogens (multiplex-PCR). The respiratory syndromatic study must also consider the diagnosis of influenza, parainfluenza, metapneumovirus, SARS-CoV-2, and RSV antigens, which increases total costs and decreases the opportunity for implementation. However, in situations of community, intra-hospital, or indoor outbreaks, it would have a recommended use due to its easy implementation [56,57].

Viral isolation through the infection of cell culture is still considered the definitive technique (gold standard) to demonstrate the presence of adenovirus in certain samples due to its high specificity (100% according to some studies) [57]. However, it is possible to obtain false negative results due to the quality of the sample, the types of cell cultures used and the incubation time, all factors that influence the sensitivity of the technique. Classically, continuous cell lines derived from epithelial carcinomas such as HEp2, HeLa and KB have been used, with similar results. Primary human embryonic kidney cultures perform better but, in practice, they are not used, given the difficulty of having this type of cell continuously available. Viral reproduction and propagation are detected by the appearance of a cytopathic effect that is recognizable without the need to stain the culture. This consists of a cellular rounding with a notable increase in refringence and a certain tendency towards a fusion between the cells. However, the confirmatory diagnosis must be made using a complementary technique. Direct IF is usually performed with monoclonal antibodies directed against common HAdV antigens, but PCR with specific primers could also be performed. Due to the complexity and special requirements for performing cell cultures, viral isolation is a technique that is difficult to apply in hospital clinical diagnosis [56].

From 2015 onwards and particularly due to the implementation of more massively used techniques associated with the SARS-CoV-2 pandemic, in Chile and worldwide, virological diagnosis was consolidated through nucleic acid amplification techniques (NAAT); prevailing over other techniques, the polymerase chain reaction (PCR) was the most used during the pandemic. In the case of HAdV, there are various commercial tests available on the market, based on Multiplex-PCR. These methods allow the recognition of various viruses in a single microbiological study with results in a couple of hours and the evaluation of respiratory syndromic symptoms in critically ill patients in the emergency department and critical patient units [57,58]. According to a study by Raty et al., who compared different diagnostic techniques for HAdV, the results obtained by PCR were much higher (94% sensitivity compared with culture as the gold standard) and comparatively superior to those of IF (46% sensitivity) and EIA (53% sensitivity), all compared with viral isolation [56,59]. Currently, the use of these molecular techniques has become widespread in large public and private medical centers in Chile. However, the cost of its implementation generates objections in public hospital management due to the high cost, around USD 100–150, so diagnostic algorithms with syndromic profiles should be developed in patients who may benefit from a study for respiratory pathogens (Multiplex-PCR) [60,61].

## 6. Application of HAdV Genotyping

HAdVs are classified into seven species (A–G) as mentioned above. Within each species, HAdV are subclassified into serotypes and/or genotypes [62]. The initial 51 serotypes were determined by neutralization assays or complement fixation assays, while genotypes 52 onwards were described by bioinformatic analysis of whole genome sequences [63,64]. In the same serotype, genomic variants could be discriminated by restriction enzyme assays [65,66]. At present, the new classification is carried out through genotyping, known as typing or “viral typing”, using whole-genome sequencing analysis. For typing purposes, the hexon gene is one of the most common targets to amplify and sequence, but fiber, penton base, and polymerase genes are also used [67,68].

Genomic variants have traditionally been detected by analyzing the electrophoretic patterns obtained after digestion with endonucleases. This methodology allowed knowing and establishing the genetic variability of the HAdV [67,69]. Today, the use of PCR amplification and sequencing has made it possible to establish phylogenetic relationships with possibilities of application in diagnosis at the clinical level. However, with this methodology it is not possible to distinguish between recombinants [69]. Various types of conventional or Realtime-PCR have been developed to establish genotypes. Some of them are genus-specific or species-specific and by subsequent sequencing, the specific genotype is established, usually directed to the hexon gene [12]. Specific genotype assays have also been described for HAdV-1 and HAdV-2 [70,71]. Other groups perform nested PCR with two pairs of amplification primers before sequencing the hexon gene [72,73]. Quantitative Realtime-PCR has been developed for types 3, 4, 7, 14, and 21 [68] and 1, 2, 5, and 6 [73]. Recently, Wu et al. described a molecular characterization based on three pairs of universal primers directed against hexon, penton base and fiber genes (viral capsid proteins), allowing typing within species B, C, D, E, F including recombinant viruses, which in recent years have been considered part of the evolution of HAdV [68]. However, this methodological approach requires post-amplification sequencing to establish the viral type [73,74].

## 7. Viral Type and Disease Association

Table 1 describes the association between HAdV types and reported diseases or infections. The most frequent types associated with pediatric respiratory infection are HAdV-1, HAdV-2 and HAdV-5 (species C) and types HAdV-3 and HAdV-7 (species B). The HAdV-4 type (species E) is also associated with respiratory infections, but mainly among military recruits, and in this group, HAdV-7 also appears, noting that HAdV-4 occurs with low frequency in the general population [9,75,76,77,78] (Table 1).

In the adult population, HAdVs that have been associated with disease in humans include species C (HAdV-1, HAdV-2 and HAdV-5), species B (HAdV-3 and HAdV-7), species E (HAdV-4) and species F (HAdV-41) predominantly in gastrointestinal pathology [78,79,80].

After primary infection, species C genotypes can establish latent infections and are capable of long-term persistence in lymphoid cells [78,80]. Therefore, asymptomatic people can shed infectious virus in the stool for many years [14,80].

## 8. HAdV Genetic Variability

In recent years, new AdV genotypes have been discovered in humans as in birds [81,82]. It has been previously demonstrated that HAdV has a higher mutation rate than other double-stranded DNA (dsDNA) such as herpes simplex virus and bacteriophage T2, but the mutation rates of dsDNA viruses are often much lower than those of RNA viruses [83]. HAdVs have an unstable genome and are subject to genetic variations, being able to have mutations by insertion, substitution or deletion of nucleotides. In addition, HAdV can also undergo genomic recombination processes, first described during the 1970s and 1980s, between different viral strains in the same individual [84,85,86]. These findings were confirmed by Dhingra et al., who demonstrated that multiple recombination events between E1 and E4 gene regions were possible as part of the evolutionary process of HAdV-C [87]. Other authors have similarly described this recombination for HAdV-16, currently poorly characterized, but it was shown recombination events with HAdV-4 type and some simian adenoviruses [88].

In 2008, Lukashev’s laboratory described that part of the phylogenetic evolution of HAdV is due to recombination between strains in the same individual [89]. They even postulated an association between the severity of the infection and the recombination between HAdV strains. Lukashev proposed that recombination processes between capsid genes (penton, hexon, fiber) could alter cell tropism. The E3 protein involved in the host’s immune control could be related to events associated with more serious infections. Many of the new genotypes described in the last decade exhibit heterotypic penton base, hexon, and fiber genes, presenting intermediate seroneutralization phenotypes (Table 2) [89,90]. In 2019, a study in China postulated that certain recombinant HAdV strains could explain why some patients had more severe respiratory symptoms [91]. It appears that the new recombinants would have synergistic mutations. Since the original HAdV strains were not particularly virulent, after recombination, the novel adenoviruses would have been associated with more severe symptoms, as described for a case of severe pneumonia in a pediatric patient in 2016 [36,91,92]. To date, only a few studies have succeeded in linking factors or molecular characteristics that make it possible to define a greater virulence of this pathogen, or that relate the viral tropism of some HAdV genotypes with parenchyma [9,64]. It has also been sought to find an association with severity in the symptoms. Therefore, it would be important to evaluate, in the future, the role of these recombinant HAdVs associated with severity in symptomatology and which molecular factors could influence increased virulence, which is discussed later [9,64].

In 2018, Cheng et al. analyzed an isolate of HAdV-55, a recombinant resulting from HAdV-B11 with renal tropism and HAdV-14 with upper airway tropism, which has been present in outbreaks of severe pneumonia in the pediatric population in China since 2006 [93]. Table 2 shows viral types identified since 2009; these HAdV emerged based on recombination of previously reported viral types. Table 2 shows the recombinant genotypes recently described in the literature, and a comparison between the ancestral penton base, hexon and fiber genes of the novel HAdV types. All these HAdV had been described in the literature as causal agents of pathologies. We observed the similarity with HAdV-14, except for the hexon gene, which are highly like HAdV-11, which would confirm the existence of recombination between them (Table 2 and Table 3) [93,94]. Table 3 shows the percent amino acid sequence identity of recombinant HAdV-55 proteins compared to other circulating human HAdV species B. Unfortunately, because of these findings, it is not feasible to carry out the identification at the level of the viral type using specific primers only for the hexon gene, since the discovery of new recombinants could not be observed. Therefore, it is recommended to evaluate the viral type by detecting the penton, hexon and fiber genes [93] (Table 3).

**Table 2 viruses-15-00330-t002:** Genbank accession number to recombinant genotypes recently described in the literature. Year of publication, origin of the recombinant gene for penton base, hexon and fiber. (Source: Human Adenovirus working group) [94].

HAdV Genotype	Name	Accession #	Year (Publication)	Penton Base	Hexon	Fiber
HAdV-B55	P14H11F14/2006/CHN	FJ643676	2009	14	11	14
HAdV-B66	P66H7F3/1987/ARG	JN860676	2012	66	7	3
HAdV-B68	P16H3F16/2004/ARG	JN860678	2011	16	3	16
HAdV-B77	P35H34F7/1985/DEU	KF268328	2013	35	34	7
HAdV-B78	P11H11F7/2013/USA	KT970441	2016	11	11	7
HAdV-C89	P89H2F2/2015/DEU	MH121097	2019	89	2	2
HAdV-C104	P1H1F2/2017/CHN	MH558113	2021	1	1	2
HAdV-C108	P1H2F2	N/A	2014	1	2	2

## 9. Association of Types, Infected Tissues and Severity of the Infection

For this review, we did not find multicenter studies with a considerable number of cases that have evaluated genotypes of the virus associated with severe disease. There are also no evaluations of large populations with surveillance of HAdV genotypes associated with heightened virulence or with description of more severe symptoms (requirement of intensive care or use of mechanical ventilation). Despite the abovementioned, we found a few studies that have shown an association between certain genotypes and infectious diseases. One of the first reports was in 2016 in China; at the Zhujiang hospital, HAdV-7 caused a longer duration of fever, greater tachypnea, dyspnea, pleural effusion, diarrhea, hepatosplenomegaly and altered states of consciousness, as well as higher rates of pneumonia, mechanical ventilation and higher mortality rates (28.6%) than other types such as HAdV-2 and HAdV-3 [95]. In 2021, a study in Turkey evaluated respiratory samples through nasopharyngeal swabs received by a national reference center (Public Health Institute of Turkey), finding an association with HAdV-F in two cases with respiratory symptoms. Previously, only F-species adenoviruses had been described in pathologies associated with intestinal tissue. HAdV species B, C, D and F were found in the overall result, with a predominance of species C. Moreover, a high percentage of lower respiratory samples with HAdV-B7 was detected, a re-emerging pathogen in that country [96]. In 2019, the diagnosis by RT-qPCR carried out on nasopharyngeal aspirates samples from a pediatric hospital in China found 5.64% of HAdV in children under 6 years old, with symptoms suggestive of pneumonia in 86.11%. of the cases, without predominance of one species over others. In this study, the most frequent pneumonia genotypes were types 2, 3 and 7. The relationship to severe disease was not analyzed with any of the genotypes described [97]. Among the few studies that have highlighted the association of genotypes and disease severity, it was found that HAdV-B3 has been identified as the pathogen causing outbreaks of severe acute respiratory diseases in Korea [98], Brazil [99] and Taiwan [100].

## 10. Relevant Characteristics of the HAdV Genome

With the use of the whole-genome sequencing (WGS) technique, viruses have been studied at the molecular level more frequently, evaluating the possible impact of proteins associated with cell tropism or disease severity [77,101,102]. WGS makes it possible to carry out a phylogenetic analysis to recognize outbreaks in progress, especially in hemato-oncology patient units or rooms for immunosuppressed patients, to take corrective measures in health institutions [102,103]. With this methodology, researchers have been able to study the HAdVs detected in outbreaks associated with health care, and the epidemiological links between hospital or in special facilities (shelters, prisons, orphanages) [16,26].

Despite the extensive development of WGS in HAdV, few proteins have been associated with severe disease, as well as the immune-regulation proteins associated with the E3 region, which codes for 6 to 9 proteins depending on the HAdV species, with different roles within the regulation of the immune response, such as downregulation of MHC I or sequestration of immune system proteins, among others [104,105,106].

Within human adenoviruses, particular proteins have been studied in HAdV-19 in search of a role in viral pathogenesis. This viral type induces the production of certain proteins. Windheim et al. reported a unique resistance system: HAdV-19 encodes a protein secreted by infected cells (sec49K), which selectively blocks the CD45 protein, one of the key regulatory molecules of leukocytes, a member of the receptor protein family tyrosine phosphatase (RPTP) [107].

## 11. Social Determinants of Health and Their Impact on HAdV

The World Health Organization (WHO) defines Social Determinants of Health (SDOH) as the circumstances under which people are born, grow, work, live and age, and the wider set of forces and systems shaping the conditions of daily life [108,109]. These conditions can be very different for different groups in a population and can lead to differences in health outcomes [10]. Although the SDOH were considered mostly for chronic pathologies such as diabetes (T2DM), obesity and high blood pressure (HBP) with SARS-CoV-2 pandemic, a preponderant role was noted in the health strategies implemented, observing that the SDOH influenced infections [10,110]. In Stockholm, Sweden the SARS-CoV-2 infection rate was 3 to 4 times higher in some low socioeconomical residential areas compared with the country average [111].

The social and economic consequences of the COVID-19 pandemic affected the entire world population, but the socioeconomic groups with limited resources were particularly affected, who experienced more deleterious consequences [112,113]. As a result of mobility restrictions, quarantines, and the perception of population risk, there was an increase in unemployment, particularly in jobs without a contract and in jobs performed by undocumented migrants, who also have more complex access to healthcare facilities [113,114]. Despite the fact that several countries have provided financial aid to reduce the basic needs of the population, their effective support has been questioned. A study carried out with data from New Zealand found that the government subsidies only prevented 6.5% of unemployment, with a heterogeneity in the statistics, since in the young adult population it could save 17.2% of the jobs in the pandemic period, while in the population over 50 years old, it was only able to protect 2.6% of jobs [115]. The risk of unemployment is higher among those who have atypical and precarious working conditions. The negative impact of unemployment on people’s health is well known and includes mental health disorders, increased alcohol and substance abuse, and family violence (domestic abuse) [116].

Overcrowding has been considered a promoter and disseminator of respiratory viruses, including HAdV [116,117,118]. HAdV has been recognized as a predominantly asymptomatic infection-producing virus. After primary infection, HAdV-C DNA can persist in a latent state in lymphoid cells, being excreted intermittently in feces for many years in immunosuppressed individuals despite being asymptomatic. Therefore, in overcrowded populations, the risk of generating outbreaks increases, especially in areas without basic environmental sanitation (non-drinking water, electricity, refrigeration) [87,119,120].

It is important to know the population at risk of acquiring HAdV, since it is generally associated with SDOH. Infants, preschoolers, older adults, and immunocompromised individuals are more likely to become infected and develop severe pneumonia [121,122,123]. Infants and preschoolers spend a lot of time in daycare centers, where outbreaks can occur due to the production of droplets and aerosols with respiratory viruses among infants, a group where control and the use of face masks are difficult [124,125]. In older adults, particularly in nursing homes, the same context can occur, generating intradomiciliary outbreaks, which can cause severe sequelae and death in these centers [126].

One of the elements that make it possible to reduce the gaps in health inequity [127] is universal access to vaccines [128,129]. In this case, there are live attenuated adenovirus vaccines for oral use developed for HAdV-4 and HAdV-7; however, they are only available for military use in some countries. Moreover, the FDA authorized them for exclusive military use in the US since 1980 [130,131]. In COVID-19, various articles have commented on inequity in access to vaccines, another social determinant in health that generates an imbalance between countries with high purchasing power and countries with lower Gross Domestic Products (GDP). The number of injected doses per population was 69 times higher in developed countries than in developing countries [132,133]. 

If an eventual HAdV pandemic is to be prevented, it would be advisable to reconsider access to a vaccine that is already available and with proven efficiency [131]. In the 2000s, there was a suspension in vaccine production due to lack of government support in the US; however, a study by Gray et al. predicted that the loss of the adenovirus vaccine would be responsible for 10,650 preventable infections, 4260 medical evaluations and 852 hospitalizations among the approximately 213,000 US Army, Navy, and Marine Corps active duty and reserve recruits enrolled each year [134]. Therefore, in the second decade of the 21st century, vaccination was reincorporated, this time for HAdV-4 and HAdV-7; this was authorized by the FDA for military personnel between 17 and 50 years of age. This measure made it possible to considerably reduce outbreaks in military recruits [135]. It is expected that in the future there will be vaccines available in the general population, given that multiple studies are developing inactivated vaccines or protein portions with demonstrated antigenicity [136,137,138]. These vaccines could reduce these already highlighted inequality gaps and eliminate some of the social determinants associated with vaccination. Therefore, universal access to those already available could be rethought, or at least in previously exposed risk groups (preschoolers, older adults and immunosuppressed people) [139,140].

## 12. Treatment

HAdV is recognized as a self-limiting pathology in infants, children, and adults, with few episodes associated with mortality [9]. However, there are recognized groups where the symptoms can evolve to severe pneumonia with requirements for critical care units, ventilatory support, and even death of patients [16,26,47]. Many drugs have been studied for the treatment of HAdV. Currently, the use of drugs remains controversial since there are no randomized prospective therapeutic trials in the medical literature [141]. 

Cidofovir (CDV) is a cytosine nucleotide analog that inhibits DNA polymerase; it has the highest in vitro activity against HAdV among the currently available antiviral agents and it is the currently preferred therapeutic agent [131,142,143,144], though it is only available intravenously [144]. Regimens (dose, frequency and period) are variable. Standard doses include 5 mg/kg every 1 or 2 weeks or 1 mg/kg twice a week, both in the pediatric and adult population. Its use is recommended in immunosuppressed patients with solid organ transplantation and hematopoietic stem cell transplantation, who have been infected with HAdV. In immunocompetent patients who require critical care units, particularly mechanical ventilation and extracorporeal membrane oxygenation (ECMO) could be used [131,143,144].

On the clinicaltrials.gov platform (accessed on 28 November 2022), when we searched with the terms “HadV” and “Treatment”, 212 research protocols were found, though only 18 have been finalized with results available for evaluation. Three addressed adenovirus as a vaccine model for the Marburg Virus and Ebola, and one for HIV. Only three studies on the platform refer to HadV treatment itself. In these three studies, they used Brancidofovir drug (BCV) (Phase II study) [145].

Brancidofovir is a new antiviral agent in early clinical trials, initially created to protect smallpox virus mutants as a bioterrorism agent [146]. CDV was its origin molecule, but due to its recognized nephrotoxicity, scientists decided to generate a prodrug with fewer side effects. In addition, it was observed that BCV was between 25 to 150 times more effective against smallpox [147]. During its development, it was observed that it had a potent activity against other double-stranded DNA (dsDNA) viruses, particularly against HAdV [148].

BCV is a lipid conjugate that covalently binds to CDV, mimicking lysophosphatidylcholine (LPC), which can use the natural pathway of LPC uptake in the small bowel [149]. BCV is a prodrug that can be used orally but, unlike typical prodrugs, remains within infected target cells. Once at the destination site, BCV is converted to CDV after cleavage of its lipid moiety, then CDV undergoes phosphorylation via the intracellular kinase pathway to form active cidofovir diphosphate (CDV-pp). CDV-pp can function as a competitive inhibitor of DNA polymerase. This drug decreases DNA synthesis, thus generating early termination of nucleotide chain elongation [149,150].

One of the phase III studies evaluated 60-day all-cause mortality in hematopoietic cell transplant recipients treated with BCV with disseminated human adenovirus [151]. Results are not yet published, but interim reports showed that in 66% of HAdV-positive patients, post-BCV treatment, no viral genome was detected in plasma, urine, feces or respiratory secretions in many cases. Mortality was 37% at day 75 of follow-up, compared with a previously reported 50%–80% mortality with other treatments such as cidofovir or without medical treatment [152]. It is expected that the next few months will provide information on the other phase III studies of this drug and its use in HAdV and other dsDNA viruses.

## 13. Adenovirus-Based Vaccines as a SARS-CoV-2 Prevention Strategy

HAdV have been used as viral vectors that produce antigenic proteins within host cells, potentially being a safe platform for the development of vaccines for multiple pathogens, for example: HIV [153,154,155], Zika virus [156], Plasmodium falciparum (malaria) [157] and SARS-CoV-2 [158,159,160,161,162]. The first antecedents were described in the mid-1990s. One of these articles was carried out by de Imler et al. demonstrating how the virus can express heterologous proteins in vitro, allowing the formation of recombinant viruses with high immunogenic power; these viruses can express a wide variety of antigens [163]. In 1995, Juillard et al. demonstrated that recombinant adenovirus vaccines with defective replication, that is, unable to produce a protein with a quaternary structure, therefore they do not generate assembly and production of new virions, had the capacity to induce a humoral response even after 6 months of administration with a single dose [164]. In this type of recombinant model, an AdV (both human and simian) was used; this AdV was incompetent to replicate in the host, which carries one or more genes of the antigenic proteins that are to be expressed, for example, SARS-CoV-2 spike protein. In this case, the recombinant virus is inoculated into specific cell cultures, so that during their replication they can express the protein on the viral surface [165,166,167]. HAdV must have the ability to interact with the host’s immune cells, allowing an antigen–antibody interaction to develop an effective cellular and humoral immune response [168].

The advances in the development of this model described in the early 1990s allowed some primary evaluations to be made in animal models in 2008 to combat eventual SARS-CoV epidemic outbreaks. This virus had caused an epidemic in Asia in previous years, with hundreds of deaths (2003) [169,170]. This study also compared the humoral immune response through inoculation of recombinant HAdV in intranasal versus intramuscular mouse vaccine models [170]. In 2012, the first tests of recombinant adenovirus vaccines with SARS-CoV were carried out. Byoung-Shik and his collaborators were pioneers in the development of an adenoviral vector that was shown to generate an immune response in mice, with an eventual protective effect through the measurement of plasmatic levels of antibodies against the SARS-CoV variants described [170]. Indubitably, these investigations allowed important scientific advances, allowing the development of vaccines for MERS-CoV [171] and Ebolavirus [172]. The safety profile of recombinant AdV vaccines has been demonstrated in a variety of clinical trials with different viral models [163,165,173,174,175,176].

These investigations allowed us to have basic knowledge, which explains that in a short period of time, new researchers were able to implement clinical studies on recombinant adenoviral vaccines against SARS-CoV-2. Countries like Russia [174], England [175], USA [176], China [177] developed their own models using different adenoviruses, both human and simian in their development [171]. Vaccine studies for the prevention of COVID-19, using an AdV vector, proved to be a safe model, with few adverse events. Various studies reported adverse events such as pain at the injection site, fatigue, and fever during the first 24 h; these symptoms were reported even less frequently in adults older than 55 years. Therefore, this demonstrated that these vaccines were safe to implement as a pharmacological model of pandemic control. A vaccine model with 25 years of research [167,177,178].

## 14. Strategies for the Prevention of HAdV

Effective control of infectious disease outbreaks is an important public health goal [179]. Without a doubt, the first strategy that a scientific team would like to develop to control a pandemic is the complete eradication of the virus. For this, wild reservoirs (zoonosis) should be considered. Without its eradication, no vaccine for human use would have an impact on viral elimination. Unfortunately, primary prevention through mass vaccination to eradicate viruses such as HAdV, influenza or SARS-CoV-2 is currently unfeasible. However, there are multiple strategies that allow control and eventually stop the spread of respiratory viruses. In 2007, Handel et al. proposed the necessary strategies for the control of new pandemics, using the SARS-CoV model [179]. They suggest that it is necessary to focus on different intervention measures, such as travel restrictions, school closures, early treatment of symptomatic cases, isolation and quarantine of positive cases and their exposed contacts and pharmacological prophylaxis for exposed patients in case of availability of drugs that decrease virus spread. Theoretically, these measures could allow us to control disease outbreaks, such as SARS, pandemic influenza, and other viral diseases [179]. Aledort et al. conducted a review of the evidence for the control of pandemic influenza through non-pharmaceutical measures. They analyzed more than 2556 scientific articles available on search engines [180]. In that review, the authors stated that it would not be convenient to base preventive strategies only on mass vaccination of the population or antiviral drugs in prophylactic therapy, since these drugs would be limited in their access, since mass production by the pharmaceutical industry is not feasible. Therefore, nations with greater purchasing power would monopolize the bid for these drugs. Accordingly, the focus of nations should be to promote control in non-pharmacological measures, such as preventive isolation of cases, quarantine of close contacts, education in the proper use of masks [181], hand hygiene and respiratory hygiene/cough etiquette [182,183]. These also include limitations on human mobility and the movement of people between cities, countries or continents and the use of a mask on planes, ships and other means of transport that make long journeys. These non-pharmacological measures have economic implications that must be evaluated. During the pandemic, there were countries with restrictions on the entry of tourists for a long pandemic period, such as China and New Zealand. In the case of China, it continued with the zero-COVID strategy until mid-2022. It is estimated that up to 1 million deaths could be avoided with this strategy of restriction, quarantine and travel limitation. The advantages of this strategy were the reduced number of deaths (2 cases from 15 May 2020 to 15 February 2022). A better balance in the distribution of resources between COVID-19 and other diseases implies a better balance between COVID-19 and other economic problems [184]. Disadvantages include the deterioration in mental health, distance in social relationships, decrease in face-to-face school activities and limitation of travel outside the territory of residence [184]. However, in countries whose main economic source is tourism, such as the Dominican Republic, they were forced to eliminate mobility restrictions early. In February 2022, all restrictions on the entry of tourists into the country were lifted to maintain the entry of foreign currency. Tourism is one of the main sources of income for the resident population [185]. Unfortunately, Aledort et al. did not find strong evidence for specific non-pharmacological measures. However, some measures did have an impact in reducing epidemic outbreaks; hand washing, cough etiquette [180,186], case investigation and contact tracing, isolation of patients [187] and rapid viral diagnosis at point of care [188,189]. In 2007, this group already recommended the use of masks in symptomatic patients and healthcare workers [190,191].

According to the review by Aledort et al. (2007), it was interesting to observe that within their recommendations they explicitly suggested not limiting physical distancing, because of the inconclusive information obtained in the reviewed articles. They suggest that the cross-evidence was limited and contradictory. Despite the propensity of influenza epidemics to spread in elementary schools, data on the effectiveness of school closures in reducing community transmission in the reviewed literature were conflicting [180,192,193]. This information can currently be refuted with more than 15,000 articles that have evaluated physical distancing in some way in the SARS-CoV-2 pandemic [194]. Glogowsky et al. studied the effect of physical distancing in Germany during COVID-19. They demonstrated that because of physical distancing there was a decrease in 84% of the cases compared to what was expected with mathematical modeling. Similar results were obtained with the decrease in deaths by 66%. These data were confirmed by various subsequent studies [195,196,197,198]. 

In 2006, Gostin et al. studied community restraint measures. They recommended evaluating the restrictions according to the evolution of community infections. At the same time, the restrictions can only be used for limited times to avoid discontent among the population [199,200]. Unlike what was documented in the SARS-CoV-2 studies, the states put sanitary measures above individual liberties. The world authorities assumed this risk to avoid a total health crisis and the collapse of health systems [201,202,203]. According to the literature evaluated in 2007, experts argued that generalized detection of travelers would be impractical and inefficient, if it is not feasible to detect asymptomatic excretion from infected patients [191]. However, 13 years later, the measure was implemented in various airports around the world. Delimited quarantines, study through Realtime-PCR or study of SARS-CoV-2 viral antigen were indicated. Scientific groups positively evaluated these measures. They described them as having a high epidemiological impact for controlling the entry of infected cases to the various countries. For example, in Toronto, all passengers over the age of 18 coming from abroad were tested. This group detected up to 2/3 of the cases that became symptomatic for COVID-19 within 14 days [203]. Other studies have shown that free tests against COVID-19 massed at airports decreased public spending on health, since they cut the infection chain upon entry to countries [204,205,206].

In summary, after two pandemics in the last 15 years (Influenza (H1N1)pdm09 and SARS-CoV-2), we can see that non-pharmacological measures are essential to control the situation in periods of maximum stress in the health system. However, there is insufficient scientific evidence to ensure which are the appropriate measures. While scientists have not reached a consensus on non-pharmacological measures, the opinion of experts and the WHO guidelines will be the alternative to follow. Among these measures, the most efficient demonstrated are (1) hand washing; (2) respiratory protection (cough etiquette and use of a mask); (3) critical analysis of the scientific literature. In 15 years, a change in scientific judgment of non-pharmacological measures has been observed, from not limiting travel, to active border surveillance, with shortened isolation in travelers, and massive studies with molecular tests for the detection of respiratory viruses in addition to using the new technologies available to trace population movement through GPS, mobile networks such as 5G technology, including applications for self-reporting of symptoms in travelers [191,205,206].

## 15. Why Is Genomic Surveillance of HAdV Important? Establishment of SARS-CoV-2 Genomic Surveillance

### 15.1. Genomic Surveillance Overview

In 1952, the World Influenza Surveillance Network was created, an initiative created by the WHO [207]. Since the 1990s, systematic surveillance of various respiratory viruses has been described in several countries. In 1992, Germany set up its first routine influenza surveillance network. Initially the samples were cultured in the Clinical Reference Laboratory for diagnostic confirmation [208]. In 2002, the virus search was through Realtime-PCR for influenza virus [209]. In parallel, the interconnected computer surveillance directed by the WHO emerged in 1998: an interconnected network of centers that centralized their data through the internet. This network allowed the detection of outbreaks in 54 centers affiliated to the software around the world [210]. From 2006 to 2009, the scientists observed that the pattern of annual cases was increasing and was not the characteristic profile of the southern hemisphere. In China, it was observed that there were two peak cases (2006, 2009). Influenza had an unusual behavior in the population. This surveillance network observed this event and alerted the WHO authorities. It was observed that in the surveillance period there was initially an increase in cases of influenza in Southeast Asia before the spread in other regions of the world [211]. 

Every time a new variant of SARS-CoV-2 emerges, it is cause for concern. The scientific community tries to understand the virological characteristics. Is this variant less or more transmissible than the original virus? Can this variant evade the immune response? Does this variant cause new symptoms? Which patients will be the most affected? Which immunocompromised people are at higher risk for severe COVID-19? To answer these questions, we need to perform the sequencing of the variants to carry out genetic studies and identify the modifications in the genome that can produce these alterations. Therefore, it is essential that worldwide surveillance of respiratory viruses continue; ideally, WGS should be carried out so that genome modifications can be recognized and associated with greater viral pathogenicity, generating alterations to the immune response in the population. This technique is proving to be important for understanding transmission patterns and potential host–virus interactions. These analyses cannot be performed with the diagnostic technique of Realtime-PCR or CRISPR (such as CRISPR Cas12a with loop-mediated isothermal amplification (LAMP)) [212,213]. CRISPR Cas12, Cas13, and Cas14 effector proteins have a unique collateral cleavage ability, which enables these tools to indiscriminately cleave surrounding nucleic acid once they bind to the target site. Cas12 recognizes dsDNA more efficiently than single-stranded DNA (ssDNA), but it still exhibits collateral activity for ssDNA. CRISPR can be combined with LAMP to simplify the detection method by visualizing the result of positive or negative samples with the naked eye, LED or UV lamps, or by observing the lateral flow strips. The combination of CRISPR Cas12 with other techniques confers specificity to the diagnosis; one of them is LAMP, allowing it to be used in viral diagnosis. Traditional methods such as PCR or RT-qPCR need expensive, bulky equipment; especially, PCR needs to either run on an agarose gel for visualization of the target DNA [213]. 

### 15.2. Surveillance in HAdV

In the case of HAdV, there are few reports on global surveillance in humans. Since 2003, the US CDC systematically initiated HAdV surveillance. During the first years, surveillance was carried out with cell culture and serotyping, currently using molecular methods. The arguments for carrying out surveillance are several: 1. To determine patterns of circulation for individual HAdV types in the US; 2. To recognize outbreaks associated with circulating HAdV types; 3. To guide the development of new diagnostic tests, therapies, and vaccines [214]. In the United States there are two institutions that complement this surveillance. The first institution led by the US Department of Defense (DoD), the Global Emerging Infection Surveillance and Response System (GEIS) established in 1997 as part of the Armed Forces Health Surveillance Branch (AFHSB) to conduct laboratory surveillance of respiratory diseases among military populations. The second institution is also part of the CDC, but with a passive surveillance system that collects data from different virological diagnostic centers in the US to carry out temporal and geographical surveillance of different respiratory viruses [215].

### 15.3. HAdV Surveillance in Wastewater

Viral surveillance in wastewater allows the evaluation of viral behavior in certain populations. The objective is to evaluate the population load of pathogens in the water, studying genetic material from the sewage system (home, hospital and industrial water). A study in Taiwan demonstrated the increase of HAdV genetic material in wastewater, and also evaluated samples obtained in sea markets near the sewage treatment (oysters and bivalve shellfish) [216]. SARS-CoV-2 surveillance in wastewater has proven to be a sensitive tool for studying spatial and temporal trends in virus circulation in the population [217,218]. Through metagenomics, the circulating virome and the behavior of different respiratory and enteric viruses and even circulating zoonotic viruses in large cities around the world were studied [218,219].

### 15.4. Surveillance and Monitoring of Zoonoses

Given the background information on genomic surveillance, research institutions and governments should consider virus surveillance in the animal kingdom a priority. In this way, the appearance of new pandemic viruses or the resurgence of old acquaintances could be recognized early. Constant evaluation of circulating zoonotic viruses is required [220]. The ECDC has been conducting avian influenza surveillance for several years, publishing reports three times a year. In the last report published in September 2022, an unusual increase in cases of highly pathogenic avian influenza is reported. 47.7 million birds were culled, with a total of 2467 reported outbreaks. In addition, two A(H5N6), two A(H9N2) and one A(H10N3) human infections were documented in China. The risk of infection is assessed as low for the general population and low to medium for persons occupationally exposed to poultry [220]. Birds are also being monitored for avian adenovirus (AdVA). In poultry, AdVA has fatal consequences, which can cause productive loss in poultry companies [221,222]. Gray et al. suggested evaluating potentially pandemic or at least epidemic viruses, including human metapneumovirus, human respirovirus 3 (formerly parainfluenza virus 3), rhinovirus C, respiratory syncytial virus, and multiple human adenovirus species. Many of these threats are now recognized as zoonotic [211]. In 2006, a novel HAdV-14 emerged in the United States and spread to other countries, causing at least 1000 illnesses and 13 deaths. While surveillance for this HAdV is sparse, available data suggest that it is spreading to at least Ireland, Canada, and China and still circulates today [222]. 

The studies related to influenza and SARS-CoV-2 surveillance, both public and private surveillance initiatives, is that it should be necessary to increase the search and sequencing of other viruses of human importance. At the same time, in the case of zoonotic viruses, animal reservoirs should be included in the study to recognize eventual pandemic events early [223,224].

## 16. Future Projections of HAdV Study in the 21st Century

The SARS-CoV-2 pandemic was possible to study with a multidisciplinary team approach. With this virus it was possible to analyze in real time the necessary requirements to avoid a disaster in the collapsed health system in developing countries [225]. In 2015, Murthy et al. exposed the deficit of ICU beds in low-income countries. These countries had an average of eight critical beds per hospital (1.5%) [226]. In 2022, the deficit of critical beds persists, and the decreased doctor–nurse–patient ratio is added in many of them [227]. Strengthening multisectoral engagement for health security was corroborated with the COVID-19 pandemic. At least the joint work of scientists, academics, health institutions, health personnel, the pharmaceutical industry, international organizations, and government institutions together is required. The WHO organized itself in a multisectoral way with its One Health initiative for the study and management of antimicrobials [228,229]. Universal access to vaccines, variations in surveillance systems, virological diagnostic tests and a combination of measures to protect vulnerable groups have managed to keep the number of SARS-CoV-2 cases at an affordable level for hospitals and health institutions. The effect of these policies allowed a decrease in the circulation of other respiratory viruses in the first months of the pandemic, reducing the overload of health systems [212,230,231,232].

To avoid a HAdV pandemic, all the groups described above are required to act together, using the tools acquired by the recent COVID-19 pandemic. The development of vaccines, genomic surveillance in sentinel centers, web pages with scientific dissemination in real time, free access to genomic sequences and multisectoral coordination help to recognize early unusual increases in cases of infection that could be important for human health and zoonosis control [119,212].

### 16.1. Unknown Hepatitis in COVID-19 Outbreak

In the immediate future, it is necessary to evaluate new evidence that allows HAdV to be confirmed or ruled out as a virus associated with cases of hepatitis. Two years after the COVID-19 pandemic, acute hepatitis of unknown etiology in children (AHUCD) began to be reported worldwide [233]. Alexander et al. reviewed a total of 22 case-control studies describing identified 1643 unknown hepatitis, with 120 children (7%) receiving liver transplants and 24 deaths (1%) [234]. The role of HAdV as a causative factor for at least some cases was theorized early and remains relevant. The initial cases identified in the UK had a perceived high positivity rate for HAdV (with 170 out of 258 [65.9%] cases having positive test results as of 4 July 2022) [235]. Gutierrez Sanchez et al. found that HAdV viremia was present in most of children with acute hepatitis of unknown cause admitted to Children’s of Alabama from 1 October 2021 to 28 February 2022, but whether HAdV was causative remains unclear. Sequencing results suggest that if HAdV was causative, this was not an outbreak driven by a single strain [236]. Kalgeri et al. conducted a retrospective study involving children referred to a single pediatric liver-transplantation center in the UK between 1 January and 11 April 2022. In these cases, with 44 young children with acute hepatitis of uncertain cause, HAdV was isolated in most of the children, but its role in the pathogenesis of this illness has not been established [237]. In all the studies evaluated, it could not be concluded that HAdV was the cause of acute hepatitis. However, all consider that a sum of factors (host, environment, biological injury agent) could be associated with liver damage. Among the associated factors, there has been an increase in the molecular diagnosis by PCR of HAdV, but it is not the only injury factor in patients with unknown hepatitis, for example, EBV, CMV, HSV, drug induced, COVID-19 vaccine, SARS-CoV-2 superantigen, environmental, etc. [238,239,240]. Based on this background, we consider that the study of hepatitis of unknown cause is an area that requires investigation by the different groups of experts worldwide in HAdV.

### 16.2. Universal Vaccination for SARS-CoV-2 with Recombinant Vaccines. What Will Be the Effect on Circulating HAdV in the Community?

The use of the vaccine in the military was previously described [241]. During the period in which HAdV vaccination was suspended, there was an unusual increase in cases, including fatal outbreaks in some US military units [242]. Several studies suggest that the use of HAdV-vectored vaccines may have little role in protective immunity, due to earlier recognition of immunity from constant lifetime exposure to these viruses [243]. In HIV vaccine studies, a risk situation was observed. In preliminary studies, there was an increased risk of acquiring HIV in the population vaccinated with HAdV-5/HIV vectored vaccines, apparently due to previous immunity against adenovirus [243]. In studies of Ebola and Influenza, they observed that using high doses of HAdV-5 viral particles, an adequate immune response was generated despite the pre-existence of antibodies in patients previously exposed to wild type HAdV-5 [244]. These data are very interesting for the study of HAdV. Since they demonstrated that previous immunity to the virus would not influence the use of HAdV as a vector for other viral proteins [245]. Could it be that the use of vectored vaccines for SARS-CoV-2 indirectly increases protection against HAdV? Or will new types simply be selected that have no antigenic response against the vaccines? To date, there are no studies that have evaluated changes in the circulation of viral types in recent years after the massive use of HAdV as a vaccine vector for other viruses [119]. This question can only be resolved with the massive use of HAdV vector vaccines. One of the safeguards in the research will be to evaluate the cross-protection that these vaccines could provide against HAdV-5 and eventually other viral types associated with species C. This could generate changes in the circulation of viral types that are not previously recognized by the immune system, generating selective pressure for new circulating viral types. By decreasing the viral load of the HAdV-5 population, a type of HAdV with global circulation and high seroprevalence in adult patients (73%) [245,246]. This could create new recombinants in previously vaccinated immunosuppressed patients [246], who, by presenting concomitant exposures of different adenoviral types, could potentially generate more aggressive recombinants. If the environmental and host conditions are adequate, recombinant pandemic precursor strains could be generated. For this reason, the future analysis will be to evaluate the epidemiological modifications in the circulation of HAdV in populations with frequent use of vector vaccines. We observed that current adenoviral vectors suppress the E1A and E1B genes, key proteins for effective viral replication. That is, the antigens will be recognized by the immune system as HAdV proteins and therefore, a subsequent immune response will be generated. We will only know the final effect with mass vaccination with adenoviral vectors and population immune-response analysis (serology) through measurement of neutralizing antibodies against HAdV-5 [166]. In the process of evaluating next-generation adenoviral vector vaccines, modifications to the viral capsid proteins are being reviewed, including alterations in structural proteins to evade immunological recognition of HAdV antigens and allow the antigenic proteins of the recombinant virus to modulate the immune response [247,248].

### 16.3. HAdV as a Vector for Other Vaccine Models

As previously explained in this article, the AdV model as a recombinant vector vaccine has been tested in multiple virus models at different stages of production. The maximum development of these recombinant vectors was achieved with COVID-19, where the vaccines have been approved by the FDA for use in pandemics [249]. This successful model of recombinant vector vaccines could be quickly implemented, if necessary, in new viral pandemics. However, its use in gene therapy for cancer treatment has also been evaluated. In the case of gene therapy, the virus could express certain antigens to be subsequently recognized by immune cells, similar to the mechanism used as a recombinant virus [250,251]. The success of vaccination with SARS-CoV-2 proposes this platform as a safe vaccine, a vaccine tested on a massive scale and with a recognized immune response against it [252,253]. Another novel investigation presents the development of vaccines for Influenza, with a recombinant HAdV-4 in a capsule for oral use. This vaccine in initial studies demonstrated circulation of neutralizing antibodies in the study subjects up to 2 years after vaccination, high levels of IgA in mucous membranes were also observed in the study period [254]. There is currently a vaccine under development with an adenoviral vector for H2N2, an avian influenza virus that circulated in humans from 1957 to 1968, a virus that caused 4 million deaths during that period [254].

### 16.4. Utility of Recognizing HAdV Genotypes. One More Step towards Genomic Surveillance

At present, the clinical characteristics associated with the severity of the infection classified by HAdV types have not been sufficiently explored. Risk factors, clinical features, circulating HAdV species, treatment, and prognosis require correct viral typing [255]. The different types of HAdV show different tissue tropisms that correlate with the clinical manifestations of the infection. The predominant rates circulating at a given time differ between countries or regions and change over time [130]. In 2016, Wang et al. one of the first groups to describe the association between recombinant HAdV-C species and symptom severity in pediatric patients [35]. In 2021, a meta-analysis conducted in China analyzed data obtained from the Pubmed and Embase library for case reports of AdVH infection. They assessed the clinical features of the disease in 228 patients. In conclusion, the authors reported that prior solid organ transplantation, hematopoietic stem cell transplantation, and hematologic malignancy were risk factors for disseminated HAdV infection. Corticosteroid use was significant for HAdV urinary tract infection. Different species were correlated with different clinical features of infection. Adenoviral types within species A, B, D, E were diagnosed more frequently in disseminated disease (with involvement of various tissues or organs) in immunosuppressed patients over F and C (*p* = 0.001), in the case of pneumonia the association was with species A, C, D, and E to a greater extent than with species B and F (*p* = 0.002). However, this study did not assess differentiation in severity between types of HAdV. Fu et al., described at Chongqing Medical University Children’s Hospital, that HAdV-7 cases were associated with most severe and fatal cases in pediatrics over HAdV-3. In the results found, there was an increase in severe pneumonia, toxic encephalopathy, leukopenia, and thrombocytopenia when compared with HAdV-3. In cell culture HAdV-7 replicates at higher levels than HAdV-3 (*p* < 0.005), increased production of C3a (*p* < 0.05) and proinflammatory cytokines (TNF-α, IL-1β, IFN-γ and IL-6) (*p* = *p* < 0.05, 0.01, and 0.001, respectively) [11]. Previously, in 2018, an association with symptom severity had already been described in HAdV-7. However, since there is no integrated genomic surveillance worldwide, it has not been possible to identify which factors affect severity in HAdV-7 or other types such as HAdV-55 [255,256]. A new HAdV-C, postulated as HAdV-104, was recently described using current typing nomenclature. It would come from the recombination of parental viruses harboring the HAdV-1 penton and hexon gene and the HAdV-2 fiber gene. This novel recombinant was discovered in a hospitalized pneumonia patient with no description of serious illness. This article reinforces what was previously evaluated. We consider recombinant HAdVs as part of the evolutionary process of HAdV-C, therefore we believe that continuous genomic surveillance with viral typing is that typing is not usually performed for the majority of diagnosed cases of HAdV respiratory infection. Only large outbreaks or cases of unusual severity have prompted detailed molecular, etiologic, or epidemiologic investigations [90,257]. We believe that, as it was implemented for Influenza, later in SARS-CoV-2, genomic surveillance should be integrated into the various respiratory viruses with pandemic potential, in the case of HAdV, typing should be included, recognition of modifications in their antigenic proteins, infectious power (viral tropism) and severity of symptoms.

### 16.5. Genomic Surveillance

The articles evaluated demonstrate the need to establish genomic surveillance worldwide; by performing complete genome sequencing, not only the viral type can be studied, but also the modifications in fundamental proteins that can predict evolutionary success, with a potential producer of a pandemic under conditions known as the epidemiological triangle (virus, environment, host). Genomic surveillance would make it possible to evaluate circulating patterns in different regions of the world, document the evolution of outbreaks, and focus diagnosis, new vaccines, and drug therapies directed to specific targets [215]. The advantage of this active surveillance of cases would allow us to anticipate unusual increases in cases associated with influenza. Mollentze et al. reported that empirical evidence suggests that generalizable signals of human infectivity might exist within viral genomes. Viruses associated with broad taxonomic groups of animal reservoirs (e.g., primates versus rodents) can be distinguished using aspects of their genome composition, including dinucleotide, codon, and amino acid biases. They proposed a particular model to predict the risk of zoonotic viruses to cause disease in humans. The model selected 645 animal-associated viruses that were excluded from training to 272 high and 41 very high-risk candidate zoonoses and showed significantly elevated predicted zoonotic risk in viruses from nonhuman primates, but not other mammalian or avian host groups. They constructed a genomic model that can retrospectively or prospectively predict the probability that viruses will be able to infect humans [258].

## 17. Conclusions

The pandemic of SARS-CoV-2 raised the question: What other respiratory viruses could cause a pandemic?

Over the past 50 years we have seen an increase in the frequency of respiratory epidemics (Asian flu, Hong Kong flu). Just since 2000, we have had four respiratory viruses associated with major epidemics (SARS-CoV, Influenza (H1N1)pdm09, MERS, SARS-CoV-2) [119]. As we have observed in this review, HAdV could perfectly generate a pandemic if the factors associated with the virus, host and environment were aligned [119]. 

For the control of respiratory viruses, it is necessary to evaluate non-pharmacological control strategies [194], have adequate distribution plans of personal protection elements to healthcare workers [259] and symptomatic patients [180]; access to rapid-result diagnostic tests at the bedside; border control, especially at airports [203]; isolation of cases and quarantine of contacts, among others, would reduce the population viral load, thus reducing transmission.

We understand that a pandemic due to RNA viruses, probably other novel coronavirus or influenza, is more likely than DNA viruses such as HAdV. However, the biotechnological use of AdV, used in recombinant vaccines or even in cancer therapies, could have yet unknown implications. Eventually, these modifications may alter ecological niches of HAdV, allow recombinants between circulating HAdV with zoonotic AdV and produce outbreaks in certain risk groups (over-crowding), such as prisons, orphanages, and military compounds. Scientific groups must continue research on possible recombinant HAdV. Zoonotic surveillance should be carried out and seroprevalence evaluated in different regions of the world.

Future projections are on the way. Genomic surveillance of the various respiratory viruses, particularly HAdV, would make it possible to evaluate the circulating types and focus vaccination strategies on limited groups [215].

Finally, we want to confirm the prominent role of HAdV in the production of recombinant vaccines, which will be very necessary if the conditions for the appearance of new pandemics continue to exist, where a rapid production of vaccines and HAdV vectorized vaccine will need to be generated. It is positioned as a known model, with proven success in the SARS-CoV-2 pandemic, but will this model not affect the circulation of other adenoviruses that are not prevalent under current ecological conditions?

## Figures and Tables

**Figure 1 viruses-15-00330-f001:**
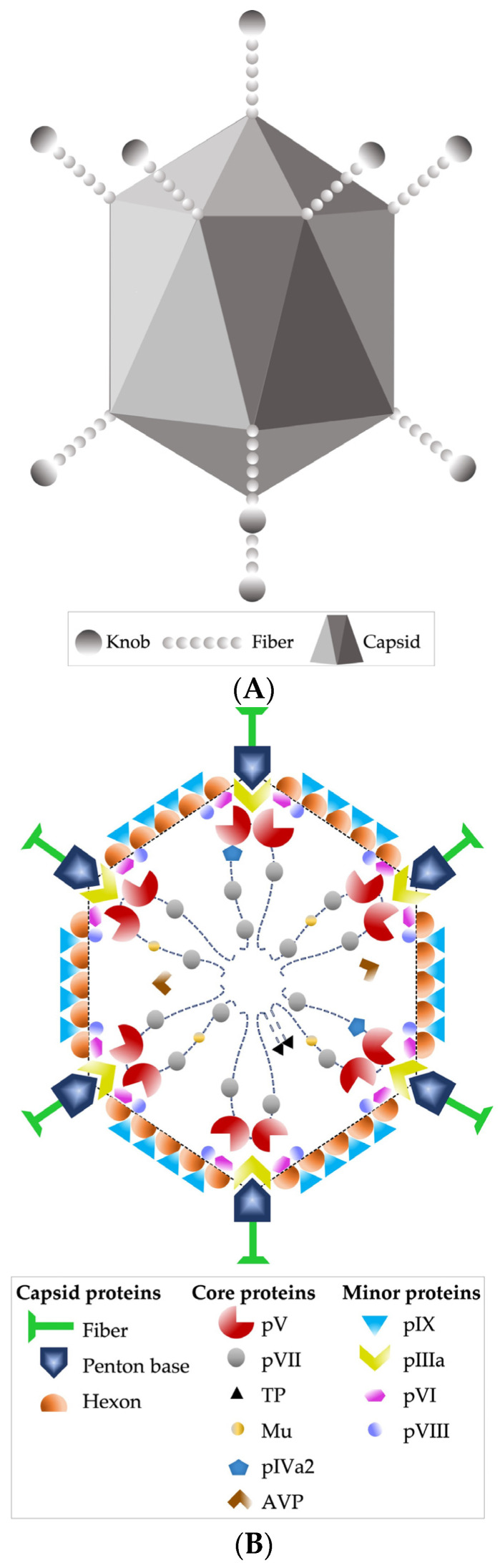
(**A**) HAdV-5 virion external schematic. (**B**) Schematic image of the DNA and protein structure within the viral capsid. Based on [33].

**Figure 2 viruses-15-00330-f002:**
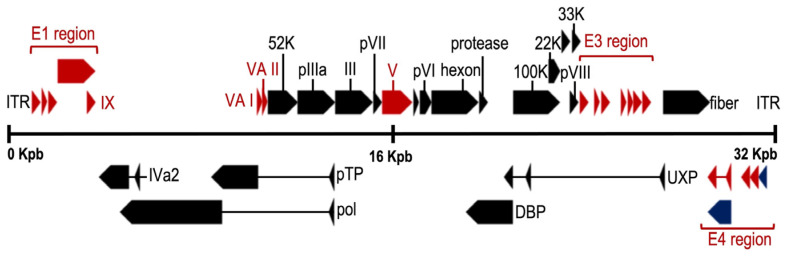
Genome organization of the mastadenovirus human adenovirus 5 (HAdV-5). Colored arrows depict genes conserved in all genera (black), present in more than one genus (blue) or restricted to mastadenoviruses (red). Rectangles mark the inverted terminal repeats. (NCBI Reference Sequence: AC_000008.1) Adapted with permission from Benkő et al. [33].

**Table 1 viruses-15-00330-t001:** Infection and HAdV Type. HAdV in the reported literature in terms of illness and severity.

			Oncogenic Potential				
Species	Hemagglutination Groups	Types	Tumors in Animals	Transformation in Cell Culture	% GC	Associated Disease	Severe or Deadly Diseases (Reported)
HAdV-A	IV (little or none)	12, 18, 31	High	Positive	46–47	Cryptic enteric infection	Unknown
HAdV-B	I (Complete for monkey erythrocyte)	3, 7, 11, 14, 16, 21, 34, 35, 50	Moderate	Positive	49–51	Conjunctivitis, Acute respiratory disease, Hemorrhagic cystitis, Central nervous system	Type 3 and 7
HAdV-C	II (Partial for rat erythrocytes)	1, 2, 5, 6	Low or none	Positive	55	Endemic infection, Respiratory symptoms	Type 5
HAdV-D	III (Complete for rat erythrocytes)	8, 9, 10, 13, 15, 17, 19, 20, 22–30, 32, 33, 36–39, 42–49, 51, 53, 54	Low or none (Mammary tumors)	Positive	55–57	Keratoconjunctivitis in immunocompromised and AIDS patients	Unknown
HAdV-E	III	4	Low or none	Positive	58	Conjunctivitis, Acute respiratory disease	Unknown
HAdV-F	III	40, 41	Unknown	Negative	51	Infantile diarrhea	Unknown
HAdV-G	Unknown	52	Unknown	Unknown	55	Gastroenteritis	Unknown

Adapted and modified from [9,76].

**Table 3 viruses-15-00330-t003:** Percent amino acid sequence identity of recombinant HAdV-55 proteins compared to other circulating human Adenovirus species B. Adapted from [93].

Protein	E1A 29.1 kDa	E1B 20 kDa	E2B DNA Polymerase	L1 pIIIa	L2 Penton Base	L3 Hexon	E2A DBP	L4 pVIII	E3 11.7 kDa	L5 Fiber	E4 ORF 6
HAdV-B35	95.8	98.3	93.2	99.5	98.6	94.4	62.9	99.1	99.1	62.1	97.7
HAdV-B34	97.7	99.4	98.7	99.3	95.3	91.3	99.4	99.1	98.1	62.2	97.7
HAdV-B14	97.7	100.0	99.7	99.8	99.5	92.2	99.8	99.6	99.1	99.1	99.3
HAdV-B11	96.6	98.3	93.0	99.3	98.4	98.4	99.2	99.1	98.1	92.3	98.0
HAdV-B7	79.4	87.8	89.9	92.7	85.3	86.6	82.6	94.3	90.6	91.1	96.7
HAdV-B3	79.0	87.2	89.9	93.0	85.5	86.3	83.8	94.3	89.6	56.7	97.3

## Data Availability

Not Applicable.

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
