# Peer review of "An Old Acquaintance: Could Adenoviruses Be Our Next Pandemic Threat?"

_viruses, 2023, doi:10.3390/v15020330_

Round 1

Reviewer 1 Report

Thanks for letting me review this draft. Here are some suggestions (major and minor) that should be addressed during preparation of a revised manuscript.

1.1       Include recent literature on the hepatitis outbreaks in children that might be caused by adenoviruses.

22.       Provide some more background on animal models to study HAdVs, see for example https://doi.org/10.3390/biology10121253.

33.       Include discussion on previous literature on the topic, like https://doi.org/10.1093/ofid/ofab078, www.pnas.org/cgi/doi/10.1073/pnas.1919176117, and https://doi.org/10.3390/v13040637, https://doi.org/10.1038/s41598-021-85849-4, https://doi.org/10.1371/journal.pbio.3001390 and possibly others…

44.       The abstract should include a sentence describing what we’ve learned from IAV and SARS as that are major parts of your main text…

55.       Line 32: What do you refer to in this sentence: “In 1953, Wallace Rowe described it for the first time in oropharyngeal lymphatic tissue.” HAdV or RSV? That should be made clear!

66.       Line 25: ”all the events” is an overstatement!

77.       Line 138: “immediate early gene” is a more common term.

88.       Line 141: It’s E1B-19K!

99.       I couldn’t find the information that you provide in Tab. 1 in the accompanying reference 9 (https://doi.org/10.51451/np.v14i1.86). Please double-check and revise if necessary.

110.   Line 272: What do you mean by “adenovirus has a higher mutation rate than another dsDNA”?

111.   Line 653: CRISPR is a diagnostic technique???

112.   Line 724: circulating AdVH?

113.   What are the author’s opinions… Could adenovirus be our next pandemic threat? Or are other (viral) infections more likely to cause the next pandemic? An educated guess of the authors would be interesting and important content of this article. I agree that adenoviruses have the potential to cause a pandemic – but is it very likely? Or are other CoVs or RNA viruses more plausible?

114.   Fig. 1 should be professionalized (use only one font/font size). Label figure panels properly. Check how to make a figure for a scientific paper.

115.   Fig. 2 seems to be copied from https://doi.org/10.1099/jgv.0.001721 à You might need to work out possible copyright issues…

116.   Author contributions: As per the International Committee of Medical Journal Editors, authorship has to be based on more than just manuscript editing/proofreading, see https://www.icmje.org/recommendations/browse/roles-and-responsibilities/defining-the-role-of-authors-and-contributors.html. I recommend to discuss this issue with all authors and come up with an acceptable solution as I am unsure if the authors stick to the journal guidelines regarding their contributions to the paper and co-authorship.

117.   English language (grammar and style) should be corrected using Grammarly or other online tools… I am aware of the effort it takes to generate clear and concise text in English. Thus, I appreciate the author’s hard work but believe that the quality of the article would significantly improve if language and style were checked and corrected/improved throughout the manuscript again.

Author Response

Dear Reviewer:

We appreciate your suggestions.

Reviewer 2 Report

This review addresses the potential for a viral pandemic resulting from Adenovirus. With the ongoing SARS-CoV-2 pandemic still ongoing this review is both timely and insightful. There are some sections that could use some reworking for clarity. In the introduction it seems that the sentence on lines 40-42 should be the concluding sentence to the opening paragraph. Same appears to the be case with sentence in lines 90-91. It seems like sentence in lines 106-108 is misplaced or could be deleted. Title for section 12 might be better mentioning HAdV based vaccines for preventing SARS-CoV-2, as written now title suggests that HAdV prevents SARS-CoV-2.

Author Response

Dear reviewer:

We appreciate your support of our review.

In the new manuscript we have accepted all your suggestions.

"This review addresses the potential for a viral pandemic resulting from Adenovirus. With the ongoing SARS-CoV-2 pandemic still ongoing this review is both timely and insightful. There are some sections that could use some reworking for clarity. In the introduction it seems that the sentence on lines 40-42 should be the concluding sentence to the opening paragraph. Same appears to the be case with sentence in lines 90-91. It seems like sentence in lines 106-108 is misplaced or could be deleted. Title for section 12 might be better mentioning HAdV based vaccines for preventing SARS-CoV-2, as written now title suggests that HAdV prevents SARS-CoV-2."

All your suggestions have been answered in the new manuscript

best regards

Reviewer 3 Report

Manuscript by Contreras et al.' gives a nice and comprehensive review of adenoviruses. In contrast to all other adenovirus reviews this one has an extra touch as it discusses the possibility that HAdV may cause the next pandemic. The manuscript covers most of the seminal studies in the adenovirus field and nicely discusses the usage of adenovirus for therapeutic purposes and also anti-adenovirus treatments. 

However, I feel that the manus is a bit too long and time-to-time loses the red line. Also, there is too much focus on SARS-CoV-2 and COVID-19. Further, the authors should pay attention to how to write out virus names: are they with capital letters or can one use acronyms? Small things that can be fixed. 

My specific comments are as following:

1)line 11: define human adenoviruses already here as (HAdV)

2) line 13: there are about 111 HAdVs described ((http://hadvwg.gmu.edu))

3)line 48: Use HAdV instead of adenoviruses

4)line 51:what do you mean "they"?

5)line 73-77: consider rephrasing & make 2 simple sentences. It is too long and the meaning hets fuzzy.

6)line 118: see my comment above ((http://hadvwg.gmu.edu)) 

7)line 140: as far I know none of the early proteins act in synthesis of proteins. This sentence is wrong! Also why the authors do not mention the essential functions of the E1B55K protein. This protein is much more studied compared to E1B19K.

8)line 223: "100-150 dollars". What dollars? USD? Hongkong dollar?

9)line 272: "another dsDNA" What is that? Feels that the sentence has not been finished.

10)line 316: use HAdV as an acronym 

11)line 371: liv????

12)line 530-575: I find it totally meaningless to have this text. Now the focus is on SARS-CoV-2 instead and it will by no means help to understand how to prevent HAdV infections. Delete it. In the worst case make 2-3 sentences of it as a summary.

13)line 653: strange sentence as it does not make sense. If you mention CRISPR (it has to be CRISPR/Cas9) then explain how it will be used!

14)line 660: HADV??? Be consistent and use HAdV!!!

15)line 701: pandemic not Pandemic!

16)line 725: what is AdVH?

17)line 757-758: totally meaningless sentence. Either delete it or rephrase it.

18)line 842: use acronym SARS-CoV-2

Figure 1A: in my 25 years of HAdV research, I have never ever seen such a wrongly drawn HAdV structure! Delete it as it is totally misleading! Use instead structure from Fig 1C.

Author Response

Dear reviewer:

Thank you very much for your comments. Your suggestions substantially improve our review. We hope that we have evaluated all the suggestions made.

Round 2

Reviewer 1 Report

Most issues have been clarified and changed… I only have minor points that should be addressed before this article can be published in Viruses.

I had these questions in my last review: What are the author’s opinions… Could adenovirus be our next pandemic threat? Or are other (viral) infections more likely to cause the next pandemic? An educated guess of the authors would be interesting and important content of this article. I agree that adenoviruses have the potential to cause a pandemic – but is it very likely? Or are other CoVs or RNA viruses more plausible?

Your answer was: “The authors of this article understand that a pandemic due to RNA viruses, probably due to other coronaviruses, possibly due to influenza, is more likely than DNA viruses such as HAdV. However, the biotechnological use of AdV, used in recombinant vaccines or even in cancer therapies, is that we do not know the future implications. Eventually, these modifications may alter ecological niches of HAdV, allow recombinants between circulating HAdV with zoonotic AdV and produce outbreaks in certain risk groups (overcrowding), such as prisons, orphanages, military compounds. We believe that we should continue investigating possible recombinant HAdV, perform zoonotic surveillance, and assess seroprevalence in different regions of the world.”

è These information/thoughts/discussion should be included in the manuscript, ideally in the conclusion paragraph.

Language issues remain… you states that “l. Once again, a full reading and revision will be carried out together with the corrections requested by the reviewers.” To me, that sounds like you’ll have someone check this manuscript again before you resubmit the next version, correct? That would be great!

Author Response

Dear Reviewer:

Thank you very much for your help.

We have corrected the manuscript with a Native speaker. We have also included the requested paragraph in the conclusion.

"We understand that a pandemic due to RNA viruses, probably other novel coronavirus or influenza is more likely than DNA viruses such as HAdV. However, the biotechnological use of AdV, used in recombinant vaccines or even in cancer therapies could have yet unknown implications. Eventually, these modifications may alter ecological niches of HAdV, allow recombinants between circulating HAdV with zoonotic AdV and produce outbreaks in certain risk groups (over-crowding), such as prisons, orphanages, military compounds. Scientific groups must continue research on possible recombinant HAdV. Zoonotic surveillance should be carried out and seroprevalence evaluated in different regions of the world."

Rev

Best regards,